# Green Tea Catechins Modulate Skeletal Development with Effects Dependent on Dose, Time, and Structure in a down Syndrome Mouse Model

**DOI:** 10.3390/nu14194167

**Published:** 2022-10-07

**Authors:** Sergi Llambrich, Rubèn González-Colom, Jens Wouters, Jorge Roldán, Sara Salassa, Kaat Wouters, Vicky Van Bulck, James Sharpe, Zsuzsanna Callaerts-Vegh, Greetje Vande Velde, Neus Martínez-Abadías

**Affiliations:** 1Biomedical MRI, Department of Imaging and Pathology, University of Leuven (KU Leuven), 3000 Leuven, Belgium; 2Departament de Biologia Evolutiva, Ecologia i Ciències Ambientals (BEECA), Facultat de Biologia, Universitat de Barcelona (UB), 08028 Barcelona, Spain; 3Laboratory of Biological Psychology, University of Leuven (KU Leuven), 3000 Leuven, Belgium; 4Center for Genomic Regulation (CRG), The Barcelona Institute of Science and Technology (BIST), 08003 Barcelona, Spain; 5Universitat Pompeu Fabra (UPF), 08003 Barcelona, Spain; 6Institució Catalana de Recerca i Estudis Avançats (ICREA), 08003 Barcelona, Spain; 7EMBL Barcelona, European Molecular Biology Laboratory, 08003 Barcelona, Spain

**Keywords:** skeletal development, Down syndrome, EGCG, µCT

## Abstract

Altered skeletal development in Down syndrome (DS) results in a brachycephalic skull, flattened face, shorter mandibular ramus, shorter limbs, and reduced bone mineral density (BMD). Our previous study showed that low doses of green tea extract enriched in epigallocatechin-3-gallate (GTE-EGCG), administered continuously from embryonic day 9 to postnatal day 29, reduced facial dysmorphologies in the Ts65Dn (TS) mouse model of DS, but high doses could exacerbate them. Here, we extended the analyses to other skeletal structures and systematically evaluated the effects of high and low doses of GTE-EGCG treatment over postnatal development in wild-type (WT) and TS mice using in vivo µCT and geometric morphometrics. TS mice developed shorter and wider faces, skulls, and mandibles, together with shorter and narrower humerus and scapula, and reduced BMD dynamically over time. Besides facial morphology, GTE-EGCG did not rescue any other skeletal phenotype in TS treated mice. In WT mice, GTE-EGCG significantly altered the shape of the skull and mandible, reduced the length and width of the long bones, and lowered the BMD. The disparate effects of GTE-EGCG depended on the dose, developmental timepoint, and anatomical structure analyzed, emphasizing the complex nature of DS and the need to further investigate the simultaneous effects of GTE-EGCG supplementation.

## 1. Introduction

Down syndrome (DS) is a congenital disorder caused by trisomy of chromosome 21 that affects 1 out of every 700–1000 live-born individuals [1]. The genetic imbalance induced by trisomy alters the development and function of multiple body systems, including the cranial and postcranial skeletal systems [2]. The skull shape is significantly altered in DS, showing a shorter, wider, and rounder shape (brachycephaly), along with a flattened face with midfacial hypoplasia and a shorter mandibular ramus [3,4,5]. Postcranial skeletal development is also altered in individuals with DS, with shorter limb long bones, shorter stature, reduced bone mineral density (BMD), increased fracture risk, and abnormal bone remodeling and growth, which may result in osteopenia and osteoporosis in adulthood [6,7,8,9,10,11,12].

Several signaling pathways regulating bone growth and development are altered in DS. Among the approximately 200 triplicated genes in chromosome 21, the dual-specificity tyrosine-(Y)-phosphorylation-regulated kinase 1A (*DYRK1A*) is a candidate gene for modulating DS skeletal phenotypes [2,13,14]. Altered *DYRK1A* expression can cooperatively destabilize the nuclear factor of activated T-cells (NFAT) regulatory circuit, leading to reduced NFATc1 (transcription factors from the ‘nuclear factor of activated T-cells activity’ family) and result in altered osteoblast differentiation and bone homeostasis [15,16]. Similarly, DYRK1A activity can affect wingless-related integration site (WNT) target genes or substrates such as sirtuin 1 (SIRT1), forkhead Box O1 (FOXO1), dickkopf WNT Signaling Pathway Inhibitor 3 (DKK3), and dishevelled segment polarity protein 1 (DVL1) and alter bone homeostasis [2]. Even though increased *DYRK1A* expression may not fully determine the constellation of skeletal anomalies associated with DS, evidence shows that regulating the signaling pathways in which *DYRK1A* is implicated can lead to rescuing effects in DS [6,17,18,19].

DYRK1A inhibitors such as epigallocatechin-3-gallate (EGCG) have been proposed for therapy in DS, because several studies have demonstrated an amelioration of brain, cognitive, and craniofacial deficits associated with DS [5,17,19,20,21,22,23,24,25]. However, the effects of EGCG as a therapeutic agent for skeletal development are not consistent, with studies showing both positive and negative effects depending on the timing and dose of treatment [26]. The administration of pure EGCG (9 mg/kg/day) in 3-week-old mice had a rescuing effect on the femoral BMD and trabecular microarchitecture [6], whereas treatment with 20 mg/kg/day of EGCG starting at week 3 for 7 weeks showed no improvements in trabecular bone and limited improvements in cortical measurements [27]. In contrast, a higher dose of EGCG (50 mg/kg/day) resulted in disturbances of the femoral strength and cortical microarchitecture [28], and a chronic daily dose of 200 mg/kg/day of pure EGCG starting at postnatal day 42 reduced cortical bone area and thickness, increased marrow area and endocortical bone surface, and decreased ultimate force and stiffness of the femora [29]. These investigations were performed under different experimental settings; therefore, the results cannot be directly compared, and the potential rescuing effect of EGCG on skeletal development in DS remains not well understood.

Moreover, most studies have tested the potential of EGCG treatments to rescue the skeletal phenotype of DS mouse models during the postnatal period, starting the administration of the treatment after reaching adulthood. However, embryonic development is critical for skeletal development, and skeletal alterations associated with DS are already present at prenatal stages [7,8,9,10,30]. Few studies to date have tested the effects of prenatal and/or perinatal EGCG treatment. McElyea et al. (2016) observed the normalizing effects in craniofacial morphology in 6-week-old Ts65Dn mice after administering 200 mg/kg of EGCG by oral gavage on embryonic day (E) 7 and E8 [13]. Our previous study indicated that a higher dose of 100 mg/kg/day of green tea extracts enriched in EGCG (GTE-EGCG) could exacerbate facial dysmorphologies, whereas a lower dose of 30 mg/kg/day significantly reduced them [5]. Considering that GTE-EGCG modulates skeletal development and may have the potential to alleviate skeletal alterations in DS, we hypothesized that earlier interventions during prenatal development could provide stronger rescuing effects, preventing the development of skeletal dysmorphologies.

Hence, in this study, we extended our previous analyses assessing the effects of continuous GTE-EGCG treatment from prenatal to adult development in two experimental settings, administering either a high or a low GTE-EGCG dose [5], to monitor the postnatal development of cranial and postcranial skeletal structures in wild-type and Ts65Dn mice in vivo. This longitudinal study analyzed, for the first time, the effects of two different doses of prenatal and postnatal GTE-EGCG supplementation on multiple skeletal structures throughout development. Unravelling the dose- and time-dependency of the effects of treatments, as well as the effects on different anatomical structures, is critical when assessing the effects of dietary GTE-EGCG supplementation for therapeutic potential. Here, we assessed whether the same treatment could have different effects depending on the developmental stage and structure to further understand the effects of GTE-EGCG on skeletal development in DS.

## 2. Materials and Methods

### 2.1. Animals, Housing, and Ethical Statement

Trisomic Ts65Dn (B6EiC3Sn-a/A-Ts (1716)65Dn) (TS) mice and euploid littermates (wildtype, WT) were obtained from our in-house breeding colony, which was established and maintained by crossing Ts65Dn females with B6EiC3SnF1/J males (refs. 001924 and 001875, the Jackson Laboratory, Bar Harbor, ME, USA). The Ts65Dn Down syndrome mouse model recapitulates the skeletal phenotype of the human condition [31,32,33] by carrying a segment with approximately 120 genes homologous to Hsa21. This segment encompasses genes upstream of Mrpl39 to the telomeric end of Mmu16 and is translocated to a small centromeric part of Mmu17 that is not syntenic to Hsa21 [34,35,36]. 

We bred 12 litters that were housed at the animal facility of KU Leuven in individually ventilated cages (40 cm long × 25 cm wide × 20 cm high) under a 12 h light/dark schedule in controlled environmental conditions of humidity (50–70%) and temperature (22 ± 2 °C) with food and water supplied ad libitum. The date of conception (E0) was determined as the day on which a vaginal plug was present. After birth, all pups were labeled with non-toxic tattoo ink (Green Tattoo Paste, Ketchum Mfg., Lake Luzerne, NY, USA) for identification throughout the longitudinal experiments. 

All procedures complied with all local, national, and European regulations and animal research: reporting of in vivo experiments (ARRIVE) guidelines, and were authorized by the Animal Ethics Committee of KU Leuven (ECD approval number P004/2016).

### 2.2. Experimental Design and Treatment

To test the genotype and GTE-EGCG treatment effects over development, the same mice were scanned at three different developmental stages: postnatal days (PD) 3, 14, and 29. We assessed the effects of continued pre- and postnatal GTE-EGCG treatment in skeletal development in two experimental settings, using the same high and low doses as in our previous study [5] (Figure 1). As EGCG crosses the placental barrier and reaches the embryo [37], GTE-EGCG treatment started prenatally at embryonic day 9 (E9) via the drinking water of the pregnant dams. After weaning, from postnatal day (PD) 21 to PD29, GTE-EGCG dissolved in water was made available to the young mice ad libitum for the entire duration of the experiment (Figure 1).

The GTE-EGCG treatment was freshly prepared every three days. Water intake was monitored in each cage and the GTE-EGCG dosages were estimated for adult mice considering that, on average, early adult mice weigh 20 g and drink 6 mL of water per day, according to our measurements. However, the received dosages were probably lower in developing embryos and pups before weaning; since previous studies have indicated that maternal plasma concentrations of catechins are about 10-fold higher than in the placenta and 50–100-fold higher than in the fetal brain [37,38]. EGCG in the milk and plasma of PD1 to PD7 pups was detected at low concentrations [22]. 

To prepare the treatments in both experimental settings, we used a green tea extract that contained 45% EGCG according to the label dosage details (Mega Green Tea Extract, Life Extension, Fort Lauderdale, FL, USA). In experimental setting 1 (high-dose treatment), we prepared a GTE-EGCG treatment at a concentration of 0.33 mg EGCG/mL. Three litters, including WT and TS mice, were treated with this high dose of GTE-EGCG, with an estimated received dosage of EGCG of 100 mg/kg/day by the adult mouse (Figure 1). In experimental setting 2 (low-dose treatment), we reduced the concentration of the GTE-EGCG treatment following a dose progression factor of 3.2 [39], resulting in a concentration of 0.09 mg EGCG/mL. Five litters were treated with this low dose of GTE-EGCG, with an expected received dosage of 30 mg/kg/day in adult mice (Figure 1). Finally, four litters were left untreated and only received water in the drinking solution. To reduce the number of experimental mice, the same WT and TS untreated mice were included as controls in both experimental settings (Figure 1).

Mice were genotyped at PD14 by PCR from ear snips, adapting the protocol in [40]. Trisomic primers, Chr17fwd-5′-GTGGCAAGAGACTCAAATTCAAC-3′ and Chr16rev-5′-TGGCTTATTATTATCAGGGCATTT-3′; and positive control primers, IMR8545-5′-AAAGTCGCTCTGAGTTGTTAT-3′, and IMG8546-5′- GAGCGGGAGAAATGGATATG-3′, were used. The following PCR cycle conditions were used: step 1: 94 °C for 2 min; step 2: 94 °C for 30 s; step 3: 55 °C for 45 s; step 4: 72 °C for 1 min (steps 2–4 repeated for 40 cycles); step 5: 72 °C for 7 min, and a 4 °C hold. PCR products were separated on 1% agarose gel.

Mice were allocated to groups according to their genotype and pharmacological intervention: WT and TS mice untreated or treated with either high- or low-dose GTE-EGCG (Figure 1). During the experiments, researchers were blinded to genotype but not to treatment, whereas during data analysis, researchers were blinded to both genotype and treatment.

Due to uncontrollable technical issues inherent to longitudinal studies, such as µCT not being operational due to technical failure on the scanning day, movement scanning artifacts, or mouse death during the experiment, sample sizes varied across groups and developmental stages. Detailed information regarding sample sizes for each experiment and analysis is provided in Appendix A. 

### 2.3. In Vivo Micro-Computed Tomography (µCT)

We performed high-resolution longitudinal in vivo µCT at postnatal days (PD) 3, 14, and 29 to visualize the skeleton over development. Mice were anesthetized by the inhalation of 1.5–2% of isoflurane (Piramal Healthycare, Morpeth, Northumberland, United Kingdom) in pure oxygen and scanned in vivo with the SkyScan 1278 (Bruker Micro-CT, Kontich, Belgium) for three minutes using the optimized parameters specified in Appendix A. Example axial, coronal, and sagittal 2D projections are provided for each stage and experimental group in Appendix A.

### 2.4. Image Data Processing

To segment the cranial and postcranial skeleton, µCT data were first reconstructed using a beam hardening correction of 10% (NRecon software, Bruker Micro-CT, Kontich, Belgium) and then loaded into Amira 2019.2 (Thermo Fisher Scientific, Waltham, MA, USA). Three-dimensional models of each skeletal structure were automatically generated by creating an isosurface based on specific thresholds for bone. Cranial and postcranial structures were analyzed separately, as explained below.

### 2.5. Shape Analysis of Craniofacial Structures

To analyze craniofacial data, we used geometric morphometrics (GM), a sophisticated body of statistical tools developed for measuring and comparing shapes with increased precision and efficiency [41,42]. We used this quantitative shape analysis to compare the morphology of craniofacial structures in WT and TS mice and to assess the effects of high- and low-dose GTE-EGCG treatments [43]. The analyses were based on the 3D coordinates of anatomical homologous landmarks recorded over the µCT 3D reconstructions of the face, skull, and mandible at postnatal days (PD) 3, 14, and 29, as defined in Appendix A. Landmarks were acquired using Amira 2019.2 (Thermo Fisher Scientific, Waltham, MA, USA), and all GM analyses were performed using MorphoJ v1.06d [44]. To avoid potential interobserver errors, craniofacial data of mice from the high- and low-dose experimental settings were analyzed separately, so that data within each experiment were consistently collected by the same observer. The same group of untreated mice was compared with treated mice from each experimental setting (Figure 1).

To extract shape information from the 3D coordinates, we performed generalized Procrustes analysis (GPA) separately for each craniofacial structure at each stage (PD3, PD14, and PD29) and for each experimental setting (high- and low-dose GTE-EGCG treatment). GPA scales, translates, and adopts a single orientation for all specimens, minimizing the influence of size [45]. To explore shape variation, we first performed principal component analysis (PCA), a data exploration technique that performs orthogonal data decomposition to reduce the high dimensionality of inter-related data and transforms the original variables into a new set of uncorrelated variables: the principal components (PCs). Then, we compared the results with those obtained from a between-group PCA (bg-PCA), an adaptation to regular PCA that maximizes separations between the groups. In this analysis, after GPA landmark superimposition, a PCA was run on the covariance matrix of the group averages (WT untreated, TS untreated, WT treated, and TS treated), and the resulting PC scores were the inputs for another PCA. The results from the PCA and bg-PCA methods were consistent, and as the main objective of this study was to identify shape differences induced by genotype and treatment, we reported the results from the bg-PCA, which represented the morphological features maximizing the differences between diagnostic groups [46,47,48]. 

Shape variation was represented by creating a morphospace based on the first two between-group principal components (bg-PCs). If slight or no dysmorphologies were associated with DS or treatment, the different diagnostic groups overlapped in the bg-PCA scatterplot, showing similar phenotypes. If there were shape differences, the different groups separated from each other in the morphospace, showing distinct phenotypes according to genotype or treatment. bg-PCA results were plotted using the ggpubr package in R [49,50].

To assess group differences, we estimated the Procrustes distances between all pairs of groups separately for each craniofacial structure and at each developmental stage. Procrustes distances were calculated as the square root of the sum of squared differences between respective landmarks in two average shapes after Procrustes superimposition. Since Procrustes distances are computed in the full multivariate data space; distances were not biased by spurious separations of groups that could result from analyses maximizing differences between groups [46,47,48]. To quantify the statistical significance of the shape differences between WT, TS, WT treated, and TS treated mice, we ran permutation tests based on the Procrustes distances and estimated the associated *p* values.

### 2.6. Shape Analysis of Postcranial Structures

Postcranial structures such as the humerus and scapula present simpler shapes in comparison with craniofacial structures; therefore, we based our comparative shape analyses on linear measurements of the length and width of the bones. To estimate these distances, we first recorded the 3D coordinates of anatomical landmarks located along the posterior–anterior axis and the medial–lateral axis of the 3D reconstructions of the humerus and scapula, as described in Appendix A. In the humerus, we estimated the length as the Euclidean distance between landmarks 1 and 2, whereas the width was the distance between landmarks 2 and 3. To estimate the length and width of the scapula, we computed the Euclidean distances between landmarks 1 and 2, and between landmarks 3 and 4 of the scapula, respectively.

### 2.7. Bone Mineral Density (BMD)

To calculate the BMD from the µCT data, we first computed the mean grey value within a volume of interest (VOI) of ten slices placed on µCT scans of the humerus and premaxilla. In the humerus, the VOI was placed below the deltoid protuberance, whereas in the premaxilla, the VOI was located at the center of the bone using CTAn software (Bruker Micro-CT, Kontich, Belgium). Then, a phantom with two known densities of hydroxyapatite (100 mg/cm^3^ and 500 mg/cm^3^) was scanned using the same settings as for the in vivo scans and a calibration line was obtained between the known hydroxyapatite densities and their corresponding CT numbers (expressed in grey values). The resulting equation was applied to calculate the BMD.

### 2.8. Statistical Analyses

We analyzed the developmental trajectories of the length and width measurements of the humerus and scapula, as well as the BMD, by fitting a mixed-effects model as implemented in GraphPad Prism 8.0. Specifically, we used the Geisser–Greenhouse correction and fitted it using restricted maximum likelihood (REML), as previously described [20]. We then performed pairwise comparisons to assess whether these variables were different between groups within each developmental stage. We compared WT vs. TS untreated mice to evaluate the genotype effect (1), WT untreated vs. WT treated mice to evaluate the treatment effect in the WT background (2), TS untreated vs. TS treated mice to evaluate the treatment effect in the trisomic background (3), WT untreated vs. TS treated mice to determine whether the treatment had a rescuing effect in trisomic mice (4), and WT treated vs. TS treated mice to evaluate whether the treatment showed different effects in the WT and trisomic background (5). During the data analyses, no mice were identified as outliers by the robust regression and outlier removal test (ROUT) [51] with a Q (maximum desired false discovery rate) of 1%.

To determine which statistical test was adequate to evaluate the statistical significance for each variable, we first assessed whether the variable was normally distributed in each group using Shapiro–Wilk tests, and then whether each pairwise comparison presented similar standard deviations (homoscedasticity) using an F-test. If one of the four mice groups was not normally distributed or one pairwise comparison was not homoscedastic, the variable was considered as not normally distributed and/or not homoscedastic. Depending on these results, we applied the tests as described in Appendix A. Finally, we performed multiple comparisons correction using the Benjamini–Hochberg procedure (Q = 5%) for each variable. All statistical analyses were performed using GraphPad Prism (v8.02, GraphPad Software, San Diego, CA, USA).

## 3. Results

### 3.1. Facial Shape in Ts65Dn Mice Partly Rescued by Low-Dose GTE-EGCG Treatment

We characterized the developmental trajectory of the craniofacial system in DS and the effect of two GTE-EGCG treatments administered from embryonic day 9 to postnatal day (PD) 29 by comparing the shapes of untreated and treated wild-type (WT) and Ts65Dn (TS) mice at three developmental stages (PD3, PD14, and PD29) and two experimental settings: high-dose GTE-EGCG treatment (experimental setting 1, exp1) and low-dose GTE-EGCG treatment (experimental setting 2, exp2) (Figure 1). In both experimental settings the same WT and TS untreated mice were included as controls, together with different mice treated with either the high dose or the low dose.

We first explored the genotype effect on facial morphology by comparing untreated WT and TS mice. The results showed that craniofacial morphological differences after birth (PD3) were subtle, because untreated WT and TS mice tended to separate in the morphospace created by the between-group principal component analysis (bg-PCA) (Figure 2A,D); however, permutation tests based on the Procrustes distances between the mean shapes of WT and TS untreated mice indicated that these facial differences did not reach statistical significance at PD3 (*P*_exp1_ = 0.1908; *P*_exp2_ = 0.6037). Facial differences became statistically significant later in postnatal development (Figure 2B,C,E,F; Appendix A) and were confirmed in both experimental settings at PD29 (*P*_exp1_ = 0.0141; *P*_exp2_ = 0.0427) (Appendix A). 

Next, to explore the GTE-EGCG treatment effects on the facial skeleton during postnatal development, treated WT and TS mice were compared between them and with untreated mice at each experimental setting. The results showed that, after birth, high-dose GTE-EGCG supplementation did not influence facial shape in WT treated mice, because they overlapped in the morphospace with WT untreated mice and both groups of WT mice were not statistically different at PD3 (Figure 2A) (*P*_exp1_ = 0.1650). However, later in development, high-dose GTE-EGCG altered the facial shape of WT treated mice, because they were displaced in the morphospace to the opposite direction of WT untreated mice (Figure 2B,C) and the permutation tests indicated that the differences between the two groups were significant at PD14 (*P*_exp1_ = 0.0005) and at PD29 (*P*_exp1_ = 0.0199). TS mice treated with high-dose GTE-EGCG were displaced from their untreated counterparts towards the range of variation in WT mice at all stages (Figure 2A–C), but the permutation tests revealed that these differences did not reach significance (Appendix A). Moreover, the permutation tests confirmed that there was no facial rescuing effect of the high-dose treatment, because TS treated mice remained significantly different from WT untreated mice at PD14 (*P*_exp1_ = 0.0141) and PD29 (*P*_exp1_ = 0.0014).

Regarding the effects of low-dose GTE-EGCG treatment, our results indicated that the treatment altered the shape of the face in WT mice since early development, because WT treated mice were displaced further away from WT untreated mice in the bg-PCAs at all stages (Figure 2D–F), and the permutation tests confirmed statistically significant differences (Appendix A). Even though TS treated and untreated mice were separated in the bg-PCA at all stages (Figure 2D–F), the permutation tests indicated that low-dose GTE-EGCG only had a significant effect in TS mice at PD14 (*P*_exp2_ = 0.0116). At this stage, TS treated mice were significantly different from TS untreated (*P*_exp2_ = 0.0116), but also from WT untreated mice (*P*_exp2_ = 0.0008), indicating that the treatment did not rescue the facial phenotype at PD14 (Figure 2E) (Appendix A). At PD29, although the facial shape of TS treated mice was not different from the TS untreated (*P*_exp2_ = 0.3125) (Figure 2F) (Appendix A), TS treated mice completely overlapped with WT untreated mice, and no significant differences were detected between them (*P*_exp2_ = 0.4596), suggesting a rescuing effect of the treatment.

Overall, the current results are consistent with our previous findings [5], indicating that the high-dose treatment had negative effects on the facial shape in WT mice and did not rescue the phenotype in TS mice, but the low-dose GTE-EGCG treatment induced partial rescuing effects in TS mice at PD29.

### 3.2. Skull Shape Associated with DS Is Not Rescued by GTE-EGCG Treatments in Ts65Dn Mice

We expanded our analysis beyond the face to other structures of the skeletal system, such as the entire skull. The results showed that untreated WT and TS mice were clearly separated in the morphospace created by the first two bg-PCs at PD3 (Figure 3A,D), PD14 (Figure 3B,E), and PD29 (Figure 3C,F). The permutation tests confirmed that genotype differences in TS mice induced significant skull dysmorphologies over development, from PD3 (*P*_exp1_ = 0.0042; *P*_exp2_ < 0.0001), to PD14 (*P*_exp1;exp2_ < 0.0001), and PD29 (*P*_exp1_ = 0.0002; *P*_exp2_ < 0.0001) (Appendix A). The bg-PC1 axis separated mice by genotype, and the morphings representing the associated shape changes showed that TS untreated mice, falling on the negative extreme of the bg-PC1 axis, presented with shorter, wider, and more globular skulls, as compared with elongated and flat skulls typically associated with WT mice, which fell on the positive extreme of bg-PC1 (Figure 3). To visualize the specific differences, we represented the skull shape differences between the extremes of bg-PCs as heatmaps (Figure 4). The heatmaps associated with bg-PC1 indicated that the genotype differences between WT and TS mice were concentrated in the cranial vault (Figure 4, first and third row, red areas).

Regarding the effect of high-dose GTE-EGCG treatment on skull shape, the results from experimental setting 1 showed that the groups were separated by treatment along bg-PC2 (Figure 3A–C). High-dose GTE-EGCG altered skull development and induced differences in WT treated mice, because they were displaced away from WT untreated mice along bg-PC2 at PD3 (Figure 3A), PD14 (Figure 3B), and PD29 (Figure 3C). Results from the permutation tests indicated that these shape changes did not reach significance in WT treated mice at PD3 (*P*_exp1_ = 0.0519), but confirmed their significance at PD14 (*P*_exp1_ < 0.0001) and PD29 (*P*_exp1_ = 0.0281). High-dose GTE-EGCG treatment also displaced TS treated mice further away from TS untreated mice across bg-PC2 (Figure 3A–C), although not significantly according to permutation tests (Appendix A). Overall, the high-dose GTE-EGCG treatment did not rescue the skull malformations associated with DS at any stage, because TS treated mice never overlapped with WT untreated mice in the bg-PCA (Figure 3A–C) and were always statistically different, as indicated by the permutation tests at every stage (*P*_exp1_ < 0.0001). The heatmaps based on bg-PC2, which separated mice based on treatment, indicated that the effects of high-dose GTE-EGCG treatment were minor at PD3 but increased with time, mostly affecting the snout and the inferior temporo-occipital region at PD29 (Figure 4, second row, red areas).

The low-dose GTE-EGCG treatment also modulated skull shape, because the groups were separated by genotype along bg-PC1 and by treatment along bg-PC2 (Figure 3D–F). The permutation tests indicated that at PD3, low-dose GTE-EGCG only affected WT treated mice (*P*_exp2_ = 0.0020), but not TS treated mice (*P*_exp2_ = 0.1430). At PD14 and PD29, low-dose GTE-EGCG treatment induced significant skull shape differences in both WT and TS treated mice (Appendix A). However, the treatment did not rescue the skull DS phenotype, because TS untreated mice were always separated from WT untreated mice in the bg-PCA and deviated WT mice from normal development, because WT treated mice were separate from WT untreated mice (Figure 3D–F). These differences were confirmed as statistically significant at all stages (*P*_exp2_ < 0.0001). According to the heatmaps, the differences induced by the low-dose GTE-EGCG treatment, visualized by green and blue areas (Figure 4, fourth row, compared with second row), were less intense than those induced by the high dose. These differences concentrated on the parietal, interparietal, and zygomatic bone at PD14, and were limited to the zygomatic bone at PD29 (Figure 4, fourth row, red areas).

### 3.3. GTE-EGCG Treatments Modulated Mandibular Shape without Rescuing Genotype Alterations

Next, we expanded our geometric morphometrics analysis to the mandible. The results showed that WT and TS untreated mice were always separated along bg-PC1 in the morphospaces representing the mandibular shape (Figure 5), indicating shape differences from birth to later developmental stages that were confirmed as significant by permutation tests at PD3 (*P*_exp1;exp2_ = 0.0002), PD14 (*P*_exp1;exp2_ < 0.0001), and PD29 (*P*_exp1;exp2_ < 0.0001) (Appendix A). The morphings representing the shape changes associated with bg-PC1 showed that TS untreated mice falling on the positive extreme had wider mandibles than WT mice, represented by the morphings on the negative extreme of the bg-PC1 (Figure 5). The heatmaps based on the comparison between the positive and negative extremes of bg-PC1 represent mandibular differences associated with the genotype, which involved localized changes in the coronoid and condyloid processes (Figure 6, first and third row, red areas).

The effects of high-dose GTE-EGCG on mandibular shape were limited. At PD3, WT treated and untreated mice were slightly separated in the bg-PCA (Figure 5A) and presented significant differences between them (*P*_exp1_ = 0.0204), whereas the differences between TS treated and untreated mice (Figure 5A) did not reach significance (*P*_exp1_ = 0.0563). At PD14 and PD29, WT and TS treated mice were separated from their untreated counterparts in the bg-PCA (Figure 5B,C), but permutation tests only confirmed significance at PD14 (*P*_exp1, WT_ < 0.0001; *P*_exp1, TS_ = 0.0007), not at PD29 (*P*_exp1, WT_ = 0.0751; *P*_exp1, TS_ = 0.2529). These results indicated a low effect of the high-dose treatment on adult mandible shape (Appendix A). The heatmaps based on the comparison between the positive and negative extremes of bg-PC2, which separated mice based on the treatment, indicated that at PD3 the differences were located around the masseteric ridge, with the high-dose GTE-EGCG treatment mainly inducing differences in the region around the mental foramen (Figure 6, second row, red areas). At PD14 and PD29, GTE-EGCG induced differences on the coronoid and condyloid processes and the masseteric ridge (Figure 6, second row, red areas). Even though high-dose GTE-EGCG induced these morphological differences, it never rescued the DS phenotype, because TS treated mice remained different from WT untreated at all stages (*P*_exp1_ < 0.0001).

In contrast, the low-dose GTE-EGCG treatment modulated the shape of the mandibles of WT and TS mice at all stages, because both treated groups were always separated from the untreated mice in the bg-PCA analyses and were significantly different from the other groups (Figure 5D–F) (Appendix A). The heatmaps based on bg-PC2 showed that low-dose GTE-EGCG mainly modulated the incisors at PD3, the masseteric ridge at PD14, and the ramus and molars at PD29 (Figure 6, fourth row, red areas). These effects, however, did not rescue the DS phenotype, because TS treated mice were consistently significantly different from WT untreated mice (*P*_exp2_ < 0.0001).

### 3.4. Upper Limb Alterations Associated with DS Not Rescued by GTE-EGCG Treatments

To further investigate the genotype and treatment effects on the skeletal system, we evaluated the length and width of the humerus and scapula, as well as the bone mineral density (BMD) of the humerus and premaxilla.

The mixed-effects model revealed that genotype differences altered the growth patterns of upper limbs. TS untreated mice presented with a significantly shorter (*P*_exp1;exp2_ = 0.0001) and narrower humerus (*P*_exp1;exp2_ < 0.0001), along with shorter and narrower scapulae (*P*_exp1;exp2_ < 0.0001) in comparison with WT untreated mice (Figure 7). These postcranial differences appeared gradually over postnatal development. Pairwise tests revealed that right after birth (PD3), the differences between TS and WT untreated mice were limited, and trisomic mice only exhibited a significantly narrower humerus (*P*_exp1_ = 0.0178; *P*_exp2_ = 0.0105). However, the morphological anomalies increased as development continued, and TS untreated mice showed a shorter and narrower humerus and scapula than WT untreated mice at PD14 and PD29 (Figure 7) (Appendix A).

High-dose GTE-EGCG significantly modified the growth trajectories of treated mice, reducing the humerus length (*P*_exp1, WT_ < 0.0001; *P*_exp1, TS_ = 0.0225) and width (*P*_exp1, WT_ = 0.0002; *P*_exp1, TS_ = 0.0043), as well as the scapula length (*P*_exp1, WT_ < 0.0001; *P*_exp1, TS_ = 0.0043) and width (*P*_exp1, WT_ < 0.0001; *P*_exp1, TS_ = 0.0004) in both WT and TS mice (Figure 7A,B,E,F). The pairwise tests did not reveal any significant high-dose treatment effects at PD3 (Appendix A), but a significant reduction in the length and width of the humerus (*P*_exp1, length_ = 0.0017; *P*_exp1, width_ = 0.0109) and scapula (*P*_exp1, length_ = 0.0002; *P*_exp1, width_ = 0.0082) in WT-treated mice at PD14 and PD29 (*P*_exp1, humerus length_ = 0.0013; *P*_exp1, humerus width_ = 0.0013; *P*_exp1, scapula length_ < 0.0001; *P*_exp1, scapula width_ = 0.0006). Regarding TS mice, the pairwise tests indicated that high-dose treatment only reduced the width of the humerus and scapula in TS treated mice at PD29 (*P*_exp1, humerus width_ = 0.0288; *P*_exp1, scapula width_ = 0.0243).

The low-dose GTE-EGCG treatment also significantly modified the growth trajectories of WT and TS treated mice. In WT mice, the low-dose GTE-EGCG significantly affected the growth trajectories of the width and length of both the humerus (*P*_exp2, width_ = 0.0060; *P*_exp2, length_ < 0.0001) and scapula (*P*_exp2, width_ < 0.0001; *P*_exp2, length_ = 0.0003). In TS mice, the low-dose treatment only affected the growth trajectory of the scapula (*P*_exp2, width_ = 0.0023; *P*_exp2, length_ = 0.0199), but not the humerus. The pairwise tests revealed that the low-dose treatment did not have any effect on the humerus shape of WT treated mice at PD3 (Appendix A), but reduced its width and length at PD14 (*P*_exp2, width_ = 0.0144; *P*_exp2, length_ = 0.0033) and PD29 (*P*_exp2, width_ = 0.0310; *P*_exp2, length_ = 0.0044). WT treated mice also exhibited a narrower scapula at PD3 (*P*_exp2_ = 0.0277), PD14 (*P*_exp2_ = 0.0343), and PD29 (*P*_exp2_ = 0.0292) and a shorter scapula only at PD14 (*P*_exp2_ = 0.0121). Regarding TS mice, the mice treated with the low-dose GTE-EGCG were only significantly different from TS untreated mice in the length of the scapula at PD3 (Appendix A). However, in contrast to TS untreated mice, the TS treated mice showed a shorter humerus and narrower and shorter scapula than WT untreated mice at this stage (*P*_exp2, humerus length_ = 0.0024; *P*_exp2, scapula width_ < 0.0001; *P*_exp2, scapula length_ = 0.0006). As development continued, TS treated mice showed a significantly shorter humerus at PD14 (*P*_exp2_ = 0.0226), narrower humerus at PD29 (*P*_exp2_ = 0.0291), and narrower and shorter scapula at PD14 (*P*_exp2, width_ = 0.0167; *P*_exp2, length_ = 0.0202) when compared with TS untreated mice (Appendix A). These results revealed that the shape of the upper limb bones was mostly unaltered at birth and both the genotype and the two doses of GTE-EGCG treatment reduced the size of the humerus and scapula throughout development.

We then compared the WT and TS mice treated with the high dose with the WT and TS mice treated with the low dose to evaluate the dose–response effects on the shape of the humerus and scapula (Figure 7I–L). We identified a dosing effect in the growth patterns of the humerus width in TS mice. The humeri of TS mice treated with the low dose were narrower than those treated with the high dose (*p* = 0.0264). Similarly, there was a dosing effect in the growth patterns of scapula length in WT mice (*p* = 0.0041), where the pairwise tests indicated that the WT mice treated with the low dose had a longer scapula at PD14 (*p* = 0.0064) and PD29 (*p* = 0.0044) than those treated with the high dose (Figure 7I–L).

Finally, because the DS phenotype involves reduced BMD [11], we calculated the BMD of the humerus and premaxilla in WT and TS mice with and without GTE-EGCG treatment. Overall, the developmental trajectories of TS and WT untreated mice were not significantly different (*P*_exp1, BMD humerus_ = 0.6618; *P*_exp2, BMD humerus_ = 0.6207; *P*_exp1;exp2, BMD premaxilla_ = 0.4177) (Figure 8) according to the mixed-effects model. However, the pairwise tests indicated that TS untreated mice had lower BMD at PD3 and PD14 for the humerus and from PD3 until PD29 for the premaxilla (Appendix A) (Figure 8). 

The high-dose GTE-EGCG treatment did not alter the developmental trajectory of the BMD of the humerus in either WT (*P*_exp1_ = 0.1836) or TS mice (*P*_exp1_ = 0.5066) (Figure 8A). However, the pairwise tests indicated that the BMD of TS treated mice was significantly lower in the humerus than WT untreated mice at PD29 (*P*_exp1_ = 0.0024), even though they were not statistically different from TS untreated mice (*P*_exp1_ = 0.4238) (Appendix A). Regarding the premaxilla, high-dose GTE-EGCG treatment altered the developmental trajectories of the BMD of the premaxilla in both WT (*P*_exp1_ < 0.0001) and TS treated mice (*P*_exp1_ = 0.0002) (Figure 8B) and reduced their BMD at PD29 (*P*_exp1, WT_ = 0.0001; *P*_exp1, TS_ = 0.0067) (Appendix A).

The low-dose GTE-EGCG treatment did not modify the developmental trajectories of the BMD of the humerus in WT (*P*_exp2_ = 0.5920) or TS untreated mice (*P*_exp2_ = 0.3608), but reduced the BMD of the humerus in WT and TS treated mice at PD14 (*P*_exp2, WT_ = 0.0035; *P*_exp2, TS_ = 0.0295) (Figure 8C). Similarly to the high-dose results, TS treated mice showed a significantly lower BMD than WT untreated mice at PD29 (*P*_exp2_ = 0.0009), even though they were not significantly different from TS untreated mice (*P*_exp2_ = 0.3557) (Appendix A). Low-dose treatment did not modulate the developmental trajectories of the BMD of the premaxilla in WT (*P*_exp2_ = 0.1489) nor TS mice (*P*_exp2_ = 0.1847), even though TS treated mice showed a significantly different developmental trajectory from WT untreated mice (*P*_exp2_ = 0.0098) (Figure 8D). The pairwise tests revealed that low-dose GTE-EGCG affected WT treated mice, reducing the BMD of the premaxilla at PD14 (*P*_exp2_ = 0.0055) and PD29 (*P*_exp2_ = 0.0001), and TS mice, reducing the BMD of the premaxilla at PD14 (*P*_exp2_ = 0.0226) (Appendix A) (Figure 8C). Overall, these results indicated that the premaxilla presented a lower BMD at birth in TS mice and that GTE-EGCG treatment had a larger effect in the premaxilla than in the humerus, reducing the BMD in both WT and TS treated mice.

When evaluating the effects of the dosing on BMD, we found different developmental trajectories between WT mice treated with the high dose and WT mice treated with the low dose for the BMD of the humerus (*p* = 0.0352) and premaxilla (*p* < 0.0001), and between TS mice treated with the high dose and the low dose for the BMD of the premaxilla (*p* = 0.0167). Remarkably, we identified that WT mice treated with the low dose had a significantly lower BMD of the humerus than the WT mice treated with the high dose at PD14 (*p* = 0.0084) (Figure 8E). The same was observed in the BMD of the premaxilla at PD3, where WT and TS mice treated with the low dose presented a significantly lower BMD than those treated with the high dose (*P*_WT_ = 0.0078; *P*_TS_ = 0.0028) (Figure 8F). At PD29, this effect was reversed, because WT mice treated with the low dose presented higher BMD values than those treated with the high dose, even though it did not reach significance (Figure 8F).

## 4. Discussion

The development of the skeletal system is altered in DS and results in delayed growth, with shorter long bones, altered craniofacial and mandibular shape, and reduced BMD [3,4,5,7,8,9,10,11,12,52,53,54]. These alterations change dynamically throughout the lifespan of individuals with DS [8,10,12], but have typically been investigated at specific timepoints using cross-sectional studies. 

The potential of green tea catechins to modulate skeletal development has been investigated in mouse models, but also at isolated stages, independently for different structures, and using different treatment regimens [5,6,13,26,28,29]. The results were inconsistent and could not elucidate the potential role of GTE-EGCG treatment on skeletal alterations associated with DS. Moreover, because each study used different techniques for bone imaging, analysis, and data interpretation, and relevant factors influencing bone development and treatment responses, such as the age and type of structure [12,55], were not consistently considered, the results from previous studies are not directly comparable [5,6,13,26,28,29]. Hence, there is limited evidence about how the same GTE-EGCG treatment simultaneously affects different skeletal structures over development. 

In this study, for the first time, we evaluated the longitudinal developmental profile of the skeletal system in the Ts65Dn mouse model of DS, assessing the effects of two doses of GTE-EGCG treatment that were administered continuously from embryonic day 9 to postnatal development, until early adulthood at PD29. We identified that TS untreated mice recapitulated the phenotype of the human condition, showing a shorter, wider, and more globular skull; wider mandible; shorter and narrower humerus and scapula; and lower BMD in the humerus and premaxilla than WT mice [4,5,7,9,10,11,12,52,54]. Our results replicated previous findings on the face, skull, and mandible morphology in newborn and adult Ts65Dn mice [3,56,57] and were consistent with the craniofacial malformations identified in other models of DS, such as TcMAC21 [58] and Dp1Tyb mice [59,60]. Our results on the long bones of the upper limb complement the findings of reduced BMD and appendicular skeletal deficiencies of Ts65Dn and Dp1Tyb mice [6,33,55,60,61]. 

In addition, our longitudinal analyses provided further insight into the development of the phenotype and the progressive appearance of the skeletal dysmorphologies. Although the skull and mandible malformations were already present in TS mice at PD3, the facial phenotype emerged later in development by PD14. Similarly, the humerus was narrower and the BMD was lower from birth, but the humerus was found to be shorter and the scapula shorter and narrower after PD14. These differences in the onset of the phenotypes reflect the dynamic nature of DS as a developmental disorder [12,55], highlighting the advantage of longitudinal experiments to pinpoint specific developmental windows where therapy may be more effective.

The dosage, onset, duration, and delivery of GTE-EGCG treatment can significantly affect the outcome response [2,12,25,27,62]. Unlike other studies that have assessed acute (GTE-)EGCG treatments [6,25,27,28], we explored the effects of a chronic GTE-EGCG treatment that started at embryonic day 9 and continued uninterruptedly until PD29. We hypothesized that combined pre- and postnatal treatment would optimize the rescuing potential of GTE-EGCG in skeletal structures. Prenatal treatment was expected to be more effective in rescuing craniofacial phenotypes, which develop early in development and minimally change after adulthood [13], whereas postcranial bones that actively remodel until late development were expected to respond more effectively to prolonged postnatal treatments. However, our longitudinal observations did not support these hypotheses. Both high- and low-dose GTE-EGCG supplementation influenced the development of the skull, mandible, long bones, and BMD, but a rescuing effect was only detected in the facial shape at PD29 with the low-dose treatment. Indeed, not only did GTE-EGCG not rescue the trisomic phenotype in the long bones, but exacerbated it, resulting in shorter bones with a lower BMD.

Our analyses emphasized that the effects of the GTE-EGCG treatments were dependent on genotype, dosing, developmental timepoint, and anatomical structure. We observed that high- and low-dose treatments affected both WT and TS mice with dose-dependent effects. High and low doses of GTE-EGCG had significantly different effects, as in the humerus width, the scapula length, and the BMD of both humerus and premaxilla. In facial shape, the low GTE-EGCG dose presented rescuing effects, whereas the high dose induced negative effects [5]. This is consistent with recent findings indicating that high doses of EGCG may impair long bone structure and strength [27] and that low concentrations of EGCG can slightly enhance osteogenesis in vivo, but higher concentrations can prevent osteogenic differentiation [63], highlighting the differential effects of the treatment depending on the dose. 

The developmental timepoint was another factor influencing the effects of the treatment. High-dose GTE-EGCG altered the mandible shape at PD3 and PD14, but not at PD29. The shapes of the faces, skulls, and upper limbs were modulated at PD14 and PD29, and the BMD of the premaxilla was reduced only at PD29. Remarkably, the low dose started showing significant effects earlier in development, modulating the shape of the face, skull, and mandible, as well as the length and width of the scapula at all stages. However, the BMD of the humerus was only reduced at PD14, and the BMD of the premaxilla and the shape of the humerus was modulated at PD14 and PD29. These results reflect the high variability of treatment responses depending on the timepoint in which the effects were analyzed, emphasizing the added value of longitudinal studies.

Finally, the different structures investigated also showed different responses to GTE-EGCG treatment. In our study, the same low dose of GTE-EGCG administered in the same mice produced anomalies in the skull, mandible, long bones, and BMD, but rescued the facial phenotype. Similarly, both treatments with GTE-EGCG produced more differences in the BMD of the premaxilla than in the BMD of the humerus, supporting the idea that different structures may react differently to the same treatment.

The differential effects of GTE-EGCG depending on the dosing, developmental timepoint, and anatomical structure should be confirmed in future studies using larger and equivalent sample sizes, but our results provide initial evidence that GTE-EGCG probably influences skeletal development through a complex mechanism of action. Further experiments assessing the complex interaction between hormones, vitamins, homocysteine levels, and gene and protein expression levels regulating growth and skeletal development are needed and will be instrumental in elucidating the mechanism of action of EGCG. Although it has been hypothesized that EGCG may rescue the DS phenotype by reducing the kinase activity of DYRK1A, this hypothesis may oversimplify a complex interaction process [29]. Further GTE-EGCG-related mechanisms could be associated with the modulation of receptor activator of nuclear factor kappa-Β ligand (RANKL) and Osteoprotegrin, as well as increasing mRNA expression of bone morphogenetic protein 2 (BMP2), RUNX family transcription factor 2 (Runx2), alkaline phosphatase (ALPL), osteonectin and osteocalcin [64,65]. Indeed, research in the Ts65Dn mouse model showed that *Dyrk1A* is not solely responsible for all the skeletal phenotypes associated with DS. Previous studies showed that the alterations of the appendicular skeleton were independent to *Dyrk1A* and NFATc [30], and that the *Dyrk1A* copy number in the osteoblasts did not affect the trabecular compartment [66]. Other genes such as ETS proto-oncogene 2 (*Ets2*), sonic hedgehog signaling molecule (*Shh*), and SRY-box transcription factor 9 (*Sox9*) may also contribute to the craniofacial and mandibular alterations in DS [67,68]. Accumulating evidence demonstrates that none of these genes are sufficient to explain the constellation of skeletal phenotypes associated with DS [69]. Indeed, no single candidate gene is responsible for the skeletal abnormalities in DS; rather, the interaction of different gene regulatory networks destabilizes the developmental pattern of the skeletal system.

Considering the potential of GTE-EGCG to ameliorate the cognitive impairment in people with DS [17,21], and to modulate the alterations of cognitive and skeletal phenotypes in mouse models [5,6,13,19,22,23,27,28,29,70], further research is needed to determine the exact mechanism of action of EGCG. Despite the challenge of accounting for the complex interaction of multiple factors such as the high genetic imbalance of DS [71,72,73], the low bioavailability of EGCG, the variate composition of the green tea extracts, and the different phenols generated after EGCG intake [25,74,75,76,77,78], basic research aiming to elucidate the mechanism of action of EGCG should be combined with observational studies evaluating the potential therapeutic effects of GTE-EGCG at various timepoints and systems. This long-term research will characterize the effects of treatments in the different structures, fine-tune the dose and timing, and fully evaluate the general safety and therapeutic potential of GTE-EGCG treatments. These longitudinal analyses on the effects of green tea catechins in health and disease are crucial because the same treatment may have disparate effects, ranging from complete or partial rescue, to no effects, or even increased alterations depending on the system, period, and dosage [5,20]. Therefore, treatments based on dietary supplements enriched in EGCG should always be taken under medical supervision.

## Figures and Tables

**Figure 1 nutrients-14-04167-f001:**
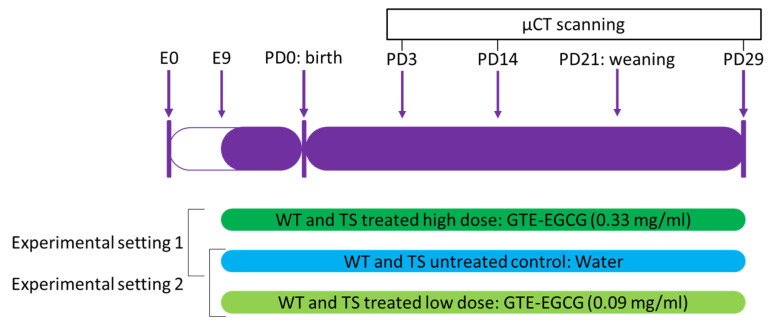
Experimental overview. Wildtype and Ts65Dn mice were scanned with in vivo µCT from birth to juvenile stages at three different stages of development (postnatal day (PD) 3, PD14, and PD29) to longitudinally follow the skeletal development. Four litters of mice were untreated and received water. Eight litters were treated with green tea extracts enriched in epigallocatechin-3-gallate (GTE-EGCG): three litters at a concentration of 0.33 mg EGCG/mL for experimental setting 1 (high-dose treatment) and five litters at a 0.09 mg EGCG/mL for experimental setting 2 (low-dose treatment), resulting in approximate doses of 100 mg/kg/day and 30 mg/kg/day, respectively.

**Figure 2 nutrients-14-04167-f002:**
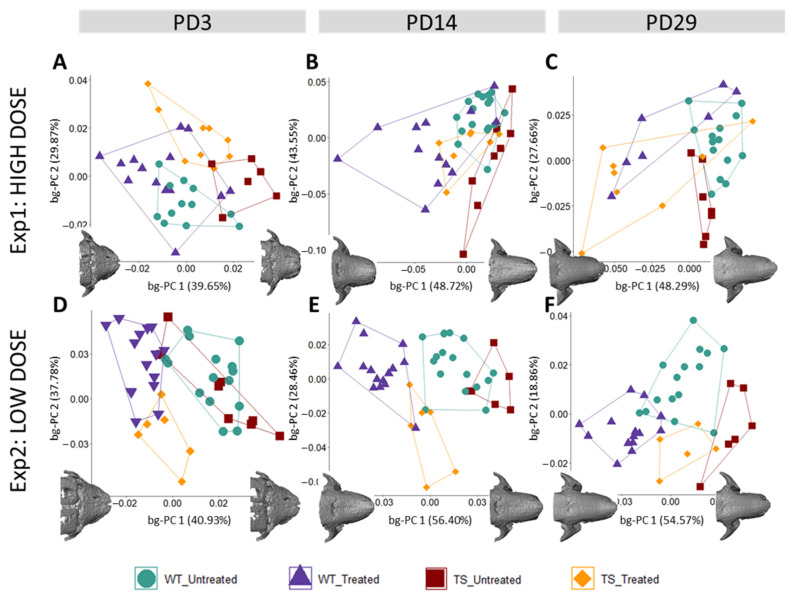
Postnatal face shape differences between wildtype and trisomic mice across development and the effects of high- and low-dose GTE-EGCG treatment. Facial shape variation as shown by a between-group principal component analysis (bg-PCA) based on the 3D coordinates of a subset of landmarks placed on the surfaces of 3D renders from in vivo µCT scans (Appendix A) at (**A**,**D**) PD3, (**B**,**E**) PD14, and (**C**,**F**) PD29. Longitudinal data from experimental setting 1 are presented on the top row, whereas data from experimental setting 2 are presented on the bottom row. All scatter plots are presented along with the morphings associated with the negative and positive extremes of the bg-PC1 axis. Notably, even though the groups of untreated mice used in experimental settings 1 and 2 were the same, the variations observed within and between groups in each experiment were not the same because the analyses included different groups of treated mice. In the context of a bg-PCA, this results in different group distributions because the total variability of the sample is different in each experiment. Sample size: Experimental setting 1: (PD3) WT = 12, TS = 6, WT treated = 14, TS treated = 9; (PD14) WT = 17, TS = 8, WT treated = 13, TS treated = 7; (PD29) WT = 14, TS = 7, WT treated = 7, TS treated = 8. Experimental setting 2: (PD3) WT = 15, TS = 8, WT treated = 15, TS treated = 6; (PD14) WT = 17, TS = 6, WT treated = 15, TS treated = 6; (PD29) WT = 15, TS = 6, WT treated = 15, TS treated = 5.

**Figure 3 nutrients-14-04167-f003:**
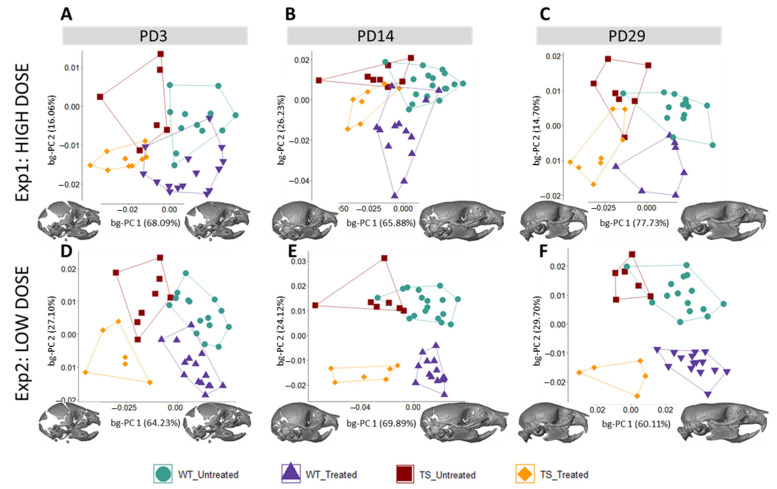
Postnatal skull shape differences between wildtype and trisomic mice across development and the effects of high- and low-dose GTE-EGCG treatment. Skull shape variation as shown by a between-group principal component analysis (bg-PCA) based on the 3D coordinates of landmarks placed on the surfaces of 3D skull reconstructions from in vivo µCT scans, as described in Appendix A at (**A**,**D**) PD3, (**B**,**E**) PD14, and (**C**,**F**) PD29. Longitudinal data from experimental setting 1 are presented at the top, whereas data from experimental setting 2 are presented at the bottom. All scatter plots are presented along with the morphings associated with the negative and positive extremes of the bg-PC1 axis. Sample size: Experimental setting 1: (PD3) WT = 12, TS = 6, WT treated = 15, TS treated = 9; (PD14) WT = 17, TS = 8, WT Treated = 13, TS Treated = 7; (PD29) WT = 14, TS = 7, WT treated = 7, TS treated = 8. Experimental setting 2: (PD3) WT = 15, TS = 8, WT treated = 15, TS treated = 6; (PD14) WT = 17, TS = 7, WT treated = 15, TS treated = 6; (PD29) WT = 15, TS = 6, WT treated = 15, TS treated = 5.

**Figure 4 nutrients-14-04167-f004:**
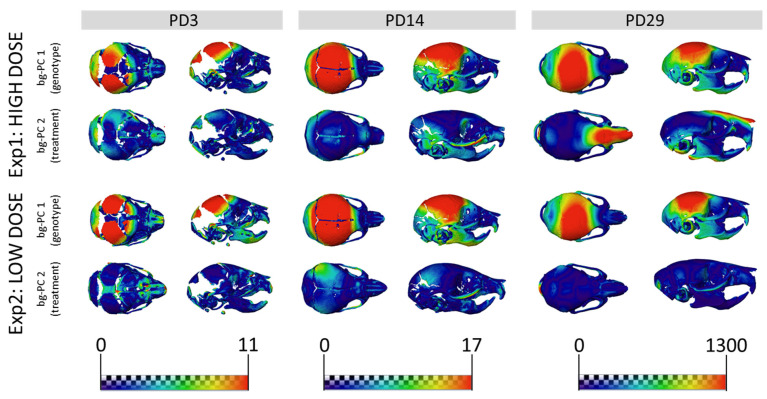
Skull morphological differences associated with genotype and treatment effects. Heatmaps showing the shape differences between the morphings on the positive and negative extremes of bg-PC1 (labeled as genotype) and the positive and negative extremes of bg-PC2 (labeled as treatment). Heatmaps are presented at PD3 (**left**), PD14 (**middle**), and PD29 (**right**) for experimental setting 1 (**top**) and experimental setting 2 (**bottom**). Red areas indicate anatomical regions with the largest shape differences. Heatmaps were derived using the shape associated with the positive extremes of bg-PC1 or bg-PC2 as the reference shape, and the shape associated with the negative extreme as the target shape.

**Figure 5 nutrients-14-04167-f005:**
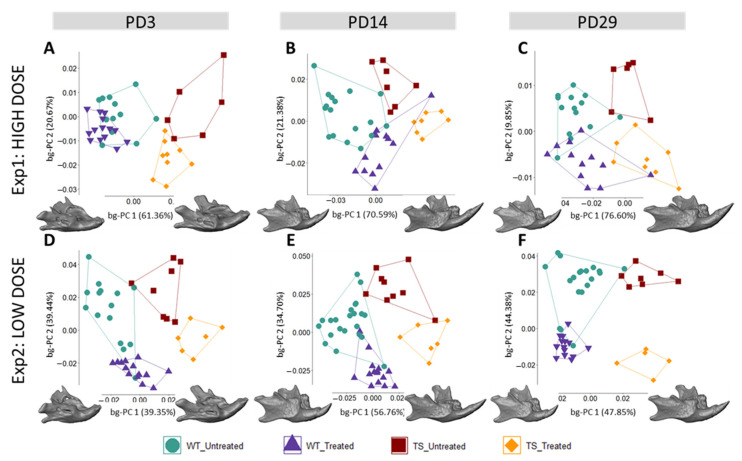
Postnatal mandible shape differences between wildtype and trisomic mice across development and high- and low-dose GTE-EGCG treatment effects. Mandible shape variation as shown by a between-group principal component analysis (bg-PCA) based on the 3D coordinates of landmarks placed on the surfaces of 3D renders from in vivo µCT scans, as described in Appendix A at (**A**,**D**) PD3, (**B**,**E**) PD14, and (**C**,**F**) PD29. Longitudinal data from experimental setting 1 are presented at the top, whereas data from experimental setting 2 are presented at the bottom. All scatter plots are presented along with the morphings associated with the negative and positive extremes of the bg-PC1 axis. Sample size: Experimental setting 1: (PD3) WT = 12, TS = 6, WT treated = 14, TS treated = 9; (PD14) WT = 17, TS = 7, WT treated = 13, TS treated = 7; (PD29) WT = 14, TS = 6, WT treated = 12, TS treated = 8. Experimental setting 2: (PD3) WT = 15, TS = 8, WT treated = 13, TS treated = 6; (PD14) WT = 22, TS = 9, WT treated = 15, TS treated = 6; (PD29) WT = 19, TS = 7, WT treated = 13, TS treated = 5.

**Figure 6 nutrients-14-04167-f006:**
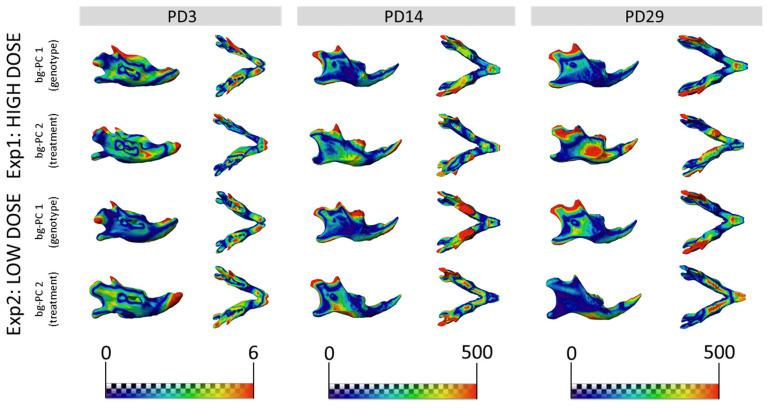
Mandibular morphological differences associated with genotype and treatment effects. Heatmaps showing the shape differences between the morphings on the positive and negative extreme of bg-PC1 (labeled as genotype) and the positive and negative extreme of bg-PC2 (labeled as treatment). Heatmaps are presented at PD3 (**left**), PD14 (**middle**), and PD29 (**right**) for experimental setting 1 (**top**) and experimental setting 2 (**bottom**). Red areas indicate anatomical regions with the largest shape differences. Heatmaps were derived using the shape associated with the positive extreme of bg-PC1 or bg-PC2 as the reference shape, and the shape associated with the negative extreme as the target shape.

**Figure 7 nutrients-14-04167-f007:**
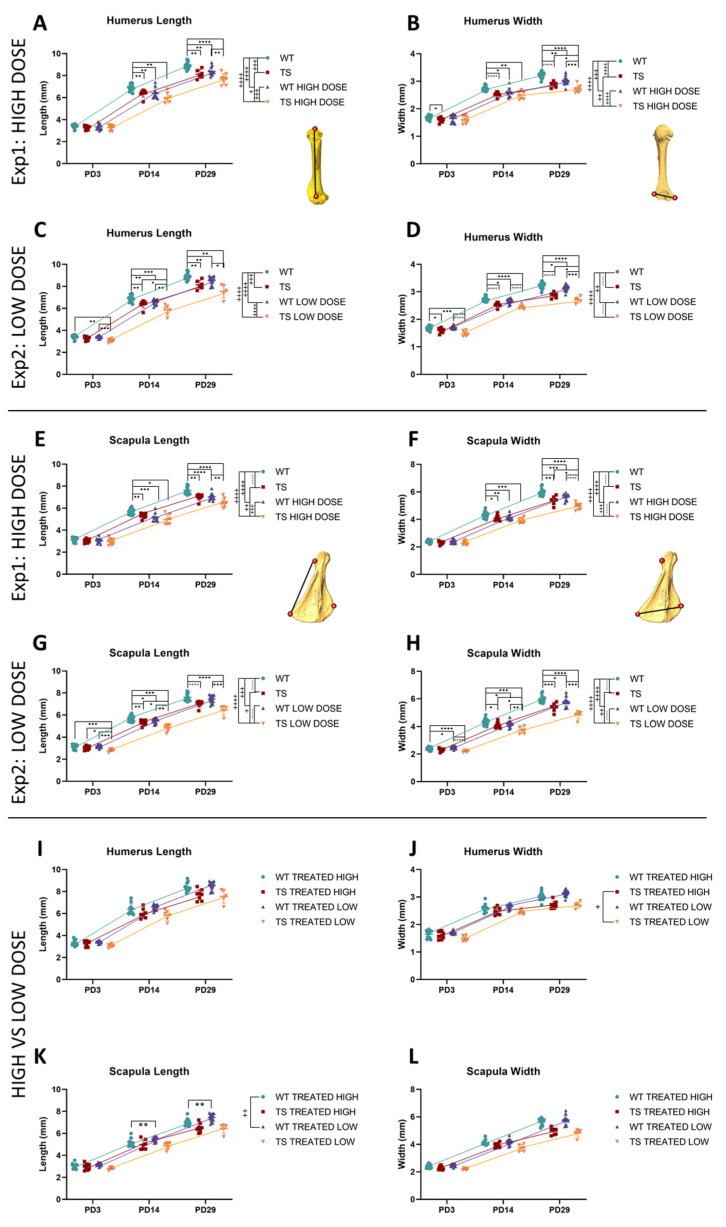
Postnatal length and width of the humerus and scapula for wildtype and trisomic mice and effects of high- and low-dose GTE-EGCG treatment across development. Euclidean distance between landmarks representing the (**A**,**C**) humerus length, (**B**,**D**) humerus width, (**E**,**G**) scapula length, and (**F**,**H**) scapula width at PD3, PD14, and PD29. (**I–L**) Comparison of the effects of the high-dose and the low-dose treatment. Data are presented as the mean +/− standard deviation. (^+^) *p* < 0.05; (^++^) *p* < 0.01; (^+++^) *p* < 0.001; (^++++^) *p* < 0.0001; Mixed-effects analysis across timepoints; (*) *p* < 0.05; (**) *p* < 0.01; (***) *p* < 0.001; (****) *p* < 0.0001; pairwise tests by timepoint. Sample size: Experimental setting 1: (PD3) WT = 15, TS = 8, WT treated = 13 (14 humerus length), TS treated = 9; (PD14) WT = 19, TS = 8, WT treated = 11, TS treated = 7; (PD29) WT = 17 (18 humerus length and width), TS = 6, WT treated = 12, TS treated = 8. Experimental setting 2: (PD3) WT = 15, TS = 8, WT treated = 15, TS treated = 6; (PD14) WT = 19, TS = 8, WT treated = 15, TS treated = 6; (PD29) WT = 17 (18 humerus length and width), TS = 6, WT treated = 15, TS treated = 5. The mice analyzed may differ across stages but represent overall ontogenetic trajectories.

**Figure 8 nutrients-14-04167-f008:**
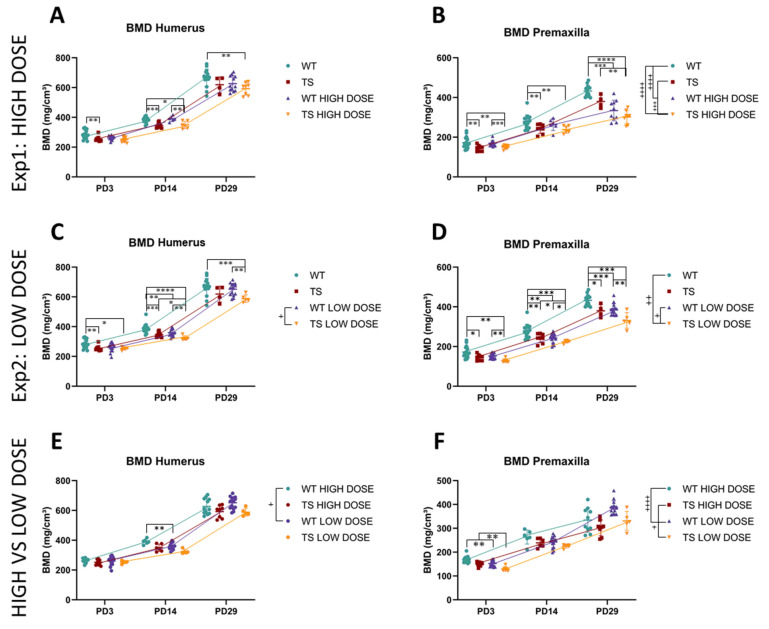
Postnatal bone mineral density of the humerus and premaxilla for wildtype and trisomic mice and the effects of high- and low-dose GTE-EGCG treatment across development. The in vivo µCT scans of the humerus and premaxilla were used to determine the (**A**,**C**) BMD of the humerus and (**B**,**D**) BMD of the premaxilla at PD3, PD14, and PD29. (**E**,**F**) Comparison of the effects of the high-dose and the low-dose treatment. Data are presented as the mean +/-standard deviation. (^+^) *p* < 0.05; (^++^) *p* < 0.01; (^+++^) *p* < 0.001; (^++++^) *p* < 0.0001; mixed-effects analysis across timepoints; (*) *p* < 0.05; (**) *p* < 0.01; (***) *p* < 0.001; (****) *p* < 0.0001; pairwise tests by timepoint. Sample size: Experimental setting 1: (PD3) WT = 18 (19 BMD premaxilla), TS = 9 (8 BMD premaxilla), WT treated = 12 (13 BMD premaxilla), TS treated = 9; (PD14) WT = 18, TS = 8, WT treated = 6, TS treated = 6; (PD29) WT = 15 (14 BMD premaxilla), TS = 4, WT treated = 12 (10 BMD premaxilla), TS treated = 8. Experimental setting 2: (PD3) WT = 18 (19 BMD premaxilla), TS = 9 (8 BMD premaxilla), WT treated = 15 (14 BMD premaxilla), TS treated = 5 (6 BMD premaxilla); (PD14) WT = 19 (18 BMD premaxilla), TS = 8, WT treated = 15, TS treated = 5 (6 BMD premaxilla); (PD29) WT = 15 (14 BMD premaxilla), TS = 4, WT treated = 15, TS treated = 5. The mice analyzed may differ across stages but represent overall ontogenetic trajectories.

## Data Availability

Data are available upon reasonable request to the corresponding authors.

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
