# Peer review of "Green Tea Catechins Modulate Skeletal Development with Effects Dependent on Dose, Time, and Structure in a down Syndrome Mouse Model"

_nutrients, 2022, doi:10.3390/nu14194167_

Round 1

Reviewer 1 Report

This isn't just an article. The requirement, high quality, and effectiveness are always present, trying to exhaustively obtain the most significant number of answers in a single report. Several themes deserve analysis in separate pieces, condensed into a single article.

The article is quite complex and writing an essay like this is not available to everyone. Not only because of the costs involved but also because this type of research is difficult and time-consuming, requiring a high technical and scientific level and rigorous coordination of the people and means involved.  

Many concepts emerged throughout the study, especially with the understanding that the effects of different drugs depend on their purity and particularly the moment in which they act.

The imaging techniques presented are fascinating and, together with the software used, allow an incredible extrapolation of data.

Despite the useful constructed figures presented, as a clinician, I would like to have had at least one or two authentic micro-CT images comparing the results of DS Mice in each stage before and after treatment.

The experience gained by these authors in this investigation will bear fruit sooner or later, as they are researching and answering new questions.

Although not conclusive, the work and dedication of these authors allow learning from the numerous doubts posed in different moments of the investigation. The experience is essential and must be transmitted.

I would suggest a modified end of the article. I understand that the authors prefer not to present conclusions. However, for the reader, a small phrase expressing the feeling of the authors would be welcome.

Reviewer 2 Report

In this manuscript, Llambrich S. et al. indicate that green tea extract, GTE-EGCG, treatment cause effective, non-effective, or adverse effect in the case of facial-, skull-, or mandibular shapes, and bone mineral densities by using Down syndrome trisomy model (Ts65Dn) and the littermate controls. This reviewer agrees that the experimental procedures are very rigid and essential to uncover the effect of high and low doses of GTE-EGCG from the specific embryonic to after-birth periods of mice. However, this reviewer also felt that the manuscript is too descriptive without any mechanical illustrations of the outcomes. This reviewer follows the journal criteria yet wishes to read the recommendations below and perform the several parts carefully.

Specific Comments;

1. Generally, it is interesting to know the dose-dependent changes but wonder why they occur. The author handles the mice until postnatal day (PD) 29, whereby they can collect the sera or plasma from each mouse. Thus, at least, the author may be able to search the several factors or hormones related to osteoblasts and osteoclasts differentiation. Why did the author calculate the enriched levels of parathyroid hormone or Vitamin K from the specimens?

2. There is a critical figure1 in methods, but it should move to the results section. This Fig. 1 is not suitable to watch in the current methods section. Thus re-format carefully in the Methods section. The methods section only can write the experimental procedures. Move to the results section if the author added any data. Similarly, in the second last paragraph in the Introduction section, which starts with ‘to characterize the altered,’ the below sentences were unsuitable for the current area. They should be created in the results or discussion section if the author wants to describe them.
